# Impact of COVID-19 pandemic social restriction measures on people with rheumatic and musculoskeletal diseases in the UK: a mixed-methods study

Toby O Smith [1,2] Pippa Belderson,[2] Jack R Dainty [2] Linda Birt [2]
Karen Durrant,[2] Jacqueline R Chipping,[2] Jordan Tsigarides,[2,3] Max Yates,[2,3]
Felix Naughton [2] Sarah Werry,[2] Caitlin Notley [2] Lee Shepstone,[2]
Alex J MacGregor [2,3]

► Prepublication history and supplemental material for this paper is available online. To view these files, please visit the journal online (http://dx.doi.org/10.1136/bmjopen-2021-048772).

¹Nuffield Department of Orthopaedics, Rheumatology and Musculoskeletal Sciences, University of Oxford, Oxford, UK
²Faculty of Medicine and Health Sciences, University of East Anglia, Norwich, UK
³Rheumatology Department, Norfolk and Norwich University Hospital, Norwich, UK

**Correspondence to**
Professor Alex J MacGregor;
a.macgregor@uea.ac.uk

## ABSTRACT

**Objectives** To determine the impact of COVID-19 pandemic social restriction measures on people with rheumatic and musculoskeletal diseases (RMDs) and to explore how people adapted to these measures over time.

**Design** Mixed-methods investigation comprising a national online longitudinal survey and embedded qualitative study.

**Setting** UK online survey and interviews with community-dwelling individuals in the East of England.

**Participants** People in the UK with RMDs were invited to participate in an online survey. A subsection of respondents were invited to participate in the embedded qualitative study.

**Primary and secondary outcome measures** The online survey, completed fortnightly over 10 weeks from April 2020 to August 2020, investigated changes in symptoms, social isolation and loneliness, resilience and optimism. Qualitative interviews were undertaken assessing participant's perspectives on changes in symptoms, exercising, managing instrumental tasks such a shopping, medication and treatment regimens and how they experienced changes in their social networks.

**Results** 703 people with RMDs completed the online survey. These people frequently reported a deterioration in symptoms as a result of COVID-19 pandemic social restrictions (52% reported increase vs 6% reported a decrease). This was significantly worse for those aged 18–60 years compared with older participants (p=0.017). The qualitative findings from 26 individuals with RMDs suggest that the greatest change in daily life was experienced by those in employment. Although some retired people reported reduced opportunity for exercise outside their homes, they did not face the many competing demands experienced by employed people and people with children at home.

**Conclusions** People with RMDs reported a deterioration in symptoms when COVID-19 pandemic social restriction measures were enforced. This was worse for working-aged people. Consideration of this at-risk group, specifically for the promotion of physical activity, changing home-working practices and awareness of healthcare

### Strengths and limitations of this study

► This study reports the impact of COVID-19 social restrictions during the first COVID-19 'lockdown' on the lived experiences of people with RMDs.

► This UK longitudinal survey of 703 people included an embedded qualitative study providing explanations on the quantitative results.

► A breadth of important health domains were examined over the initial COVID-19 10-week social restriction period, including pain, social isolation and loneliness, resilience and optimism.

► We only recruited UK residents so while the principles may apply to countries experiencing social restrictions due to COVID-19, results may not be generalisable beyond the UK.

provision is important, as social restrictions continue in the UK.

## INTRODUCTION

Rheumatic and musculoskeletal disease (RMDs) are a major cause of disability and reduced quality of life worldwide.[1] People with RMDs experience pain, joint stiffness, fatigue and muscle weakness with resultant physical disability. These symptoms are frequently managed with a combination of physical activity and medications, which for some people include immunosuppressive drugs.

In March 2020, the WHO declared the outbreak of a novel coronavirus disease SARs-CoV-2 (COVID-19) to be a pandemic. On 16 March 2020, the UK government instructed all people aged 70 years and over to self-isolate in their homes for up to 4 months. This was extended to all people on 23 March for an initial 3-week period. People who were taking immunosuppressant medications

or with medical comorbidities were instructed to maintain strict self-isolation principles (often referred to as shielding) for an extended period of at least 12 weeks.

At the outset, there was concern that people with RMDs may be particularly vulnerable to the unintended consequences of these 'lockdown' measures. They are recognised to be at increased risk of social isolation, through poor social support,[2] the frequent presence of other medical conditions[3 4] and functional disability.[5] Social isolation itself can affect levels of physical disability, psychological distress and pain in RMDs.[6] The loss of exercise opportunities, a key component of self-management programmes, risked increasing levels of disability and reducing mental well-being. COVID-19 pandemic measures also significantly impacted on routine healthcare access and National Health Service services.[7 8] Prior to this study, the resilience of this group to these enforced changes was unknown, as were likely long-term impacts.

This study explores the impact of COVID-19 pandemic social restriction measures on people with RMDs and determines how people adapted to these measures over time. Our aim was to provide insight into the challenges and potential impacts of those most at risk and to make recommendations as to how best to support people with RMDs throughout further social restriction measures and beyond.

## METHODS AND ANALYSIS
### Design
Mixed-methods investigation comprising a national online longitudinal survey and embedded qualitative study. The study is reported in accordance with the Strengthening the Reporting of Observational Studies in Epidemiology statement for reporting observational studies[9] and the Consolidated criteria for Reporting Qualitative research statement for reporting qualitative research.[10]

### Survey study
#### Cohort and recruitment
We invited participants who self-reported a diagnosis of chronic (3 months or longer) RMDs (disease of the bone, joint or muscle) and who lived in the UK to participate in an online survey.

We used three approaches to recruit potential participants. First, we approached patient-based organisations (Versus Arthritis, Arthritis Action, Royal Osteoporosis Society, National Rheumatoid Arthritis Society, National Ankylosing Spondylitis Society, Fibromyalgia UK, Age UK, PainConcern and Scope). All agreed to disseminate an invitation to the survey via their online patient forum or as an email. Second, we contacted members from an inception cohort of 1396 people with inflammatory arthritis – the Norfolk Arthritis Register (NOAR).[11] Finally, we used research team members' Twitter accounts to publicise the study.

#### Data collection
An online survey administered through the Qualtrics platform (Qualtrics XM, https://www.qualtrics.com/uk/)

was provided to all participants. The survey is presented in online supplemental file 1. Participant consent was obtained through the platform before proceeding to the study survey. The survey collected information on:
- ► Age, gender, ethnicity, duration of symptoms, type of RMD and medical comorbidities.
- ► Clinical disease activity.
- ► Changes in medication use, access to healthcare support, physical activity, disease symptoms since COVID-19 lockdown.
- ► Clinical Health Assessment Questionnaire[12] to assess disability and symptom status including: pain, fatigue, sleep and anxiety.
- ► The Lubben Social Network Scale 6[13] – short assessment of social isolation.
- ► University of California, Los Angeles three-item loneliness scale.[14]
- ► Six-item Brief Resilience Scale.[15]
- ► Revised Life Orientation Test.[16]

We opened round 1 (baseline) recruitment from 28 April 2020 to 27 May 2020. We sent tweets weekly during round 1 recruitment period. For those who completed round 1, we sent the same survey at 2-week intervals to investigate how symptoms, social isolation, loneliness, resilience and optimism changed over time (rounds 2–6). We sent a reminder email to participants who missed a subsequent round. The final survey was completed on 20 August 2020.

#### Data analysis
The analysis first addressed the perceived changes in symptoms since the start of lockdown in response to the question, '*How have your symptoms been since the current COVID-19 measures started?*'. The responses were scored as decreased, stable or increased. The influence of possible explanatory variables on the categories of symptom response was assessed through $\chi^2$ tests. Data were analysed using R (R Core Team[17]).

A second analysis examined whether there were any prospective changes over the 10-week observation period in variables (pain, fatigue, anxiety, sleep, social networks, loneliness, resilience and optimism) that could be sensitive to change as the UK Government guidelines on social restriction measures altered (online supplemental file 2). There was a focus on differences in the two age groups (18–60 years and 60+ years) with the cut-off chosen to approximate to 'working' (n=351) and 'non-working' groups (n=351). Mixed models were used to analyse repeated data from the same participants. Time was included as a fixed-effect and 'participant ID' as a random effect. All analyses were conducted in R using the 'clmm' function in the 'ordinal' package[18] for ordinal response variables or the 'lmer' function in the 'lme4' package[19] for continuous response variables.

#### Qualitative study
The qualitative study was an embedded interview study that aimed to explore the subjective experiences of

people with RMDs during COVID-19 pandemic social restrictions.

## Sample and recruitment

Survey participants recruited through NOAR were offered the option to take part in interviews; 137 expressed an interest. We designed a purposive sampling strategy, recruiting to achieve variation in age and gender, recognising that rheumatoid arthritis (RA) is more common in women and people aged over 40 years. A sample size of 25–30 was planned to support data collection in the window of guidance on self-isolation. This sample target was sufficient to enable robust themes to be developed. All those contacted agreed to take part.

## Data collection

The interview topic guide (online supplemental file 3) took a narrative approach, starting with open questions on everyday life and then focused on the participant's perspective on changes in symptoms, exercising, managing instrumental tasks such as shopping, medication and treatment regimens, and how they experienced changes in their social networks. Questions were reviewed by clinical and lay colleagues for clarity and relevance. Participants were approached by two researchers (LB and PB) over the telephone or email and received study information 72 hours prior to interview. All gave recorded verbal or written consent before the interview commenced. All interviews were conducted by two experienced qualitative researchers (LB and PB) over the telephone due to COVID-19 restrictions and were audio recorded for transcription. Interviews lasted between 30 and 90 min; average duration was 45 min.

## Data analysis

Data were analysed using a thematic analysis approach guided by the six-steps outlined by Braun and Clarke.[20] As a first step, interviews were transcribed and read to aid familiarisation with the data. Two researchers (LB and PB) coded the data in NVivo V.12. During coding, descriptive categories were identified relating to symptom attribution, changes to everyday life and well-being, narrative of vulnerability and risk management. Frequent researcher and multidisciplinary meetings led to interpretive themes.[21] To enhance the dependability and confirmability of the results, peer and participants validation was completed.[22] Emerging themes and categories were shared across the research team and rheumatology health practitioners. Interview participants were sent a summary of key themes over email, followed by a phone call to each seeking feedback during August–September 2020. There was strong resonance between the themes and participants' experiences.

The qualitative data were reviewed alongside the survey data to provide detailed explanations of experience of symptoms and well-being since restrictive measures commenced. Through this, an interpretative synthesis of results from each research methodology was presented using a mixed-methods approach.

## Patient and public involvement

Patient involvement began during protocol development stage of the study protocol and continued throughout. A patient-member (SW) provided her personal insights on the questions posed in the survey and qualitative topic guide. She provided advice on the interpretation of the survey and interview analysis. Our patient member assisted in the preparation of the final paper. She will continue to support dissemination through the preparation of public documents and social media outputs to share the results to wider patient communities.

# RESULTS

## Survey results

### Baseline assessment

In total, 703 respondents (574 females, 126 males, three others or non-binary) were recruited (262 from NOAR and 441 from the wider UK base). Their characteristics are summarised in table 1. The majority of respondents were aged 51–70 years (508/703; 72.3%) and residing in England (660/703; 93.9%). A range of inflammatory and non-inflammatory diseases were represented with RA (44.2%) and osteoarthritis (21.6%) reported most frequently. There were no important differences in the characteristics or responses in the NOAR and non-NOAR identified RA patients, and their data were pooled for analysis.

Table 2 illustrates respondent's symptoms at baseline compared with before the imposition of the COVID-19 pandemic social restriction measures. In total, 365 (52%; 95% CI 49% to 56%) reported an increase in symptoms related to their RMD, 287 (41%; 95% CI 37% to 45%) reported that they had stayed the same and 45 (6%; 95% CI 5% to 9%) reported reduction in symptoms.

Age had a significant influence on the change in symptoms, with a higher proportion of younger respondents (age ranged 18–60 years) reporting an increase in symptoms (p=0.017). As might be expected, people who experienced an increase in symptoms were more likely to have changed their medication (p<0.001) and have needed to access medical advice (p<0.001). Physical activity also varied according to the change in reported symptoms (p<0.001): those who reported that their symptoms had increased also reported a decreased level of physical activity (table 2).

### Follow-up assessment

Approximately 25%–30% of the participants were lost to attrition over the course of the study. At the second timepoint (week=2), there were 525 participants, at week 4 there were 506, at week 6 there were 540, at week 8 there were 521 and at week 10 there were 491 participants. Online supplemental file 4 compares the characteristics of participants at baseline and week 10 and

**Table 1** Respondent characteristics and responses to health provision access from online survey at baseline

| Characteristics | | Frequency (%) |
|---|---|---|
| N | | 703 |
| Gender | Female | 574 (82.0) |
| | Male | 126 (18.0) |
| | Non-binary | 1 (0.1) |
| | Prefer to self-describe | 1 (0.1) |
| | Prefer not to say | 1 (0.1) |
| Age (years) | 18–30 | 21 (3.0) |
| | 31–40 | 44 (6.3) |
| | 41–50 | 113 (16.1) |
| | 51–60 | 173 (24.6) |
| | 61–70 | 203 (28.9) |
| | 71–80 | 132 (18.8) |
| | 80+ | 16 (2.3) |
| Region of respondent | England | 660 (93.9) |
| | Scotland | 21 (3.0) |
| | Wales | 19 (2.7) |
| | Northern Ireland | 2 (0.3) |
| Ethnic group | White | 682 (97.8) |
| | Mixed/multiple ethnic group | 4 (0.6) |
| | Asian | 7 (1.0) |
| | Black, African and Caribbean | 3 (0.4) |
| | Arab | 1 (0.1) |
| RMD diagnosis | Rheumatoid arthritis | 311 (44.2) |
| | Osteoarthritis | 152 (21.6) |
| | Mechanical low back pain | 70 (9.9) |
| | Fibromyalgia | 68 (9.6) |
| | Psoriatic arthritis | 64 (9.1) |
| | Inflammatory polyarthritis | 63 (8.9) |
| | Hypermobility | 40 (5.6) |
| | Specific RMD diagnosis not reported | 33 (4.7) |
| | Connective tissue disease (eg, lupus, scleroderma and myositis) | 25 (3.5) |
| | Ankylosing spondylitis | 22 (3.1) |
| | Osteoporosis | 18 (2.5) |
| | Polymyalgia rheumatica | 10 (1.4) |
| | Ligament/tendon injury/bursitis | 9 (1.3) |
| | Neck pain | 6 (0.8) |
| | Gout | 5 (0.7) |
| | Other | 63 (8.9) |
| How have your RMD symptoms been since the COVID-19 measures started? | Decreased | 45 (6.5) |
| | Stayed the same | 287 (41.1) |
| | Increased | 365 (52.4) |

Continued

**Table 1** Continued

| Characteristics | | Frequency (%) |
|---|---|---|
| Pain (scale: 0–10) | Mean (SD) | 4.8 (2.6) |
| General health (scale: 0–10) | Mean (SD) | 4.1 (2.4) |
| Total Lubben Social Network Score (scale: 0–30) | Mean (SD) | 14.8 (5.5) |
| Total UCLA Loneliness (scale: 3–9) | Mean (SD) | 5.1 (2.0) |
| Difficulty accessing medication | Yes | 82 (11.7) |
| | No | 616 (88.3) |
| Required someone to help access medications | Yes | 309 (44.3) |
| | No | 389 (55.7) |
| Changed medications since COVID-19 outbreak | Yes | 103 (14.8) |
| | No | 595 (85.2) |
| Required to seek advice from a health professional on condition | Yes | 252 (36.1) |
| | No | 446 (63.9) |
| Who did you contact | General practitioner | 158 (22.3) |
| | Practice nurse/GP nurse practitioner | 23 (3.2) |
| | Rheumatology department | 97 (13.7) |
| | Physiotherapy or occupational therapist | 16 (2.3) |
| | Pharmacist | 16 (2.3) |
| | Hospital department (non-RMD) | 10 (1.4) |
| | A&E | 3 (0.4) |
| | Private chiropractor, osteopath or massage therapist | 1 (0.1) |
| | Royal Osteoporosis Society | 1 (0.1) |
| | Endocrinology department | 7 (1.0) |
| | Pain clinic | 2 (0.3) |
| | Counsellor or health psychologist | 2 (0.3) |
| | Massage therapist | 1 (0.1) |
| | Podiatrist | 1 (0.1) |
| | Nutritionist | 1 (0.1) |
| | NHS 111 | 2 (0.3) |
| How easy has it been to get advice? (scale: 0–10) | (Mean (SD) value (scale 0–10) | 4.8 (3.3) |

A&E, accident and emergency; JIA, juvenile inflammatory arthritis; NHS, National Health Service; RMD, rheumatic and musculoskeletal diseases; UCLA, University of California Los Angeles Loneliness Score.

**Table 2** Association at baseline between rheumatic and musculoskeletal disease (RMD) symptoms and selected participant characteristics and questionnaire responses

| | 'How have your symptoms been since the current COVID-19 measures started?' | | | $\chi^2$ test of association with outcome (symptoms) |
|---|---|---|---|---|
| | **Decreased** | **Stable** | **Increased** | **P value** |
| n (%) | 45 (6) | 287 (41) | 365 (52) | |
| Age group (years) | | | | 0.017 |
| 18–60 | 22 (6) | 125 (36) | 200 (58) | |
| 60 plus | 33 (7) | 162 (46) | 165 (47) | |
| Gender | | | | 0.110 |
| Male | 8 (7) | 60 (49) | 54 (44) | |
| Female | 37 (6) | 224 (39) | 311 (54) | |
| RMD diagnosis* | | | | <0.001 |
| RA | 17 (5) | 155 (48) | 149 (46) | |
| IA | 4 (10) | 24 (60) | 12 (30) | |
| PsA | 4 (6) | 22 (34) | 38 (59) | |
| Other | 19 (7) | 83 (31) | 166 (62) | |
| Situation | | | | 0.880 |
| At home | 24 (7) | 141 (40) | 191 (54) | |
| Self-isolating | 8 (7) | 49 (40) | 64 (53) | |
| Shielding | 13 (6) | 97 (44) | 110 (50) | |
| Difficulty accessing medication? | | | | 0.103 |
| Yes | 4 (5) | 26 (32) | 52 (63) | |
| No | 41 (7) | 261 (42) | 313 (51) | |
| Change medication? | | | | <0.001 |
| Yes | 9 (9) | 21 (20) | 73 (71) | |
| No | 36 (6) | 266 (45) | 292 (49) | |
| Consult health professional? | | | | <0.001 |
| Yes | 15 (6) | 64 (25) | 172 (69) | |
| No | 30 (7) | 223 (50) | 193 (43) | |
| Physical activity | | | | <0.001 |
| Decreased | 31 (7) | 141 (30) | 302 (64) | |
| Same | 3 (3) | 89 (75) | 26 (22) | |
| Increased | 11 (10) | 57 (54) | 37 (35) | |

Data are frequency (%) unless stated otherwise.
*Other=mainly osteoarthritis (55%) but also including any diagnosis that was not RA, IA or PsA (see table 1).
IA, inflammatory arthritis; PsA, psoriatic arthritis; RA, Rheumatoid arthritis; RMD, rheumatic and musculoskeletal diseases.

indicates that they were similar. Figures 1 and 2 illustrate the changes in responses over 10 weeks stratified by age (the strongest predictor of the increased level of symptoms). For all variables examined (symptoms, supportive social networks, loneliness, resilience and optimism), the younger age groups (18–60 years) fared worse than the older group (60 years plus) across all time points. Levels of optimism among the younger age groups fell into the range that is classified as 'low'. For the 18–60 years age group, there was a significant improvement in the levels of pain (p<0.001) and sleep (p<0.001), over the 10-week period, while anxiety levels (p=0.769) and fatigue (p=0.920) stayed the same. The 60 years plus age group had significant improvements in pain (p<0.001) and fatigue (p=0.002) but sleep (p=0.080) and anxiety (p=0.610) stayed the same over the 10 weeks (figure 1). The size of any improvements was marginal in both age groups. In contrast, feelings of social isolation intensified (18–60 years: p<0.001; 60 years plus: p<0.001), and levels of resilience (18–60 years: p<0.001; 60 years plus: p<0.001) and optimism were significantly reduced (18–60 years: p=0.008; 60 years plus: p=0.009; figure 2) but the

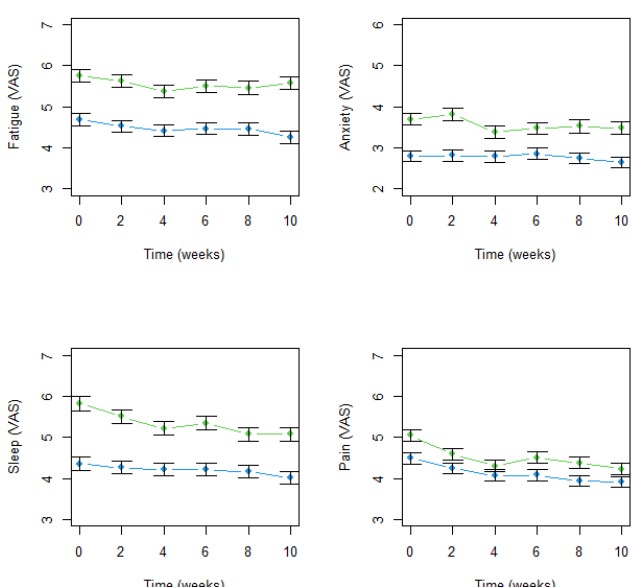

**Figure 1** Change in symptoms from baseline over the 10-week follow-up interval. Green points/lines indicate 18–60 age group; Blue points/lines indicate 60+ age group; bar widths represent one standard error of the mean. VAS, visual analogue scale.

effect sizes were, again, small and unlikely to be clinically relevant.

## Qualitative study

Interviews were undertaken with 26 participants between 27 May 2020 to 19 June 2020. The sample consisted of 9

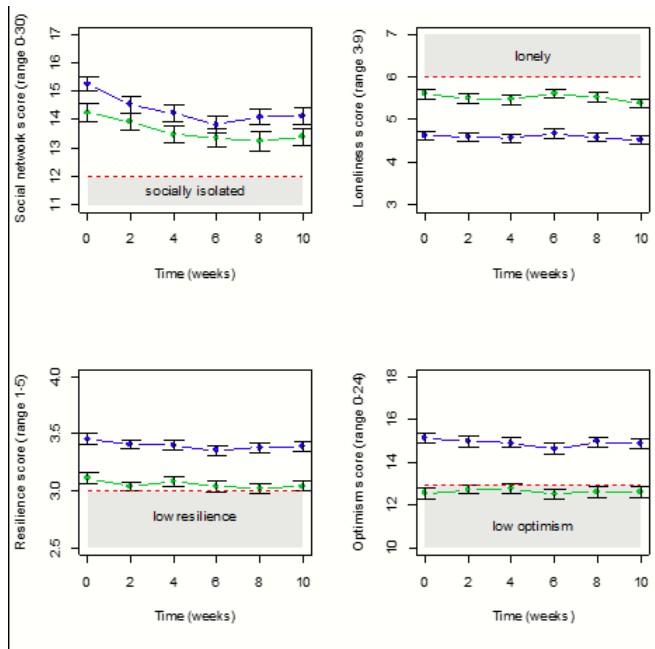

**Figure 2** Change in behaviour outcomes from baseline over the 10-week follow-up interval. Green points/lines indicate 18–60 age group; Blue points/lines indicate 60+ age group; bar widths represent one standard error of the mean.

| Table 3 | Interview subsample characteristics (n=26) | |
|---|---|---|
| **Characteristics** | | **Frequency** |
| Gender | Female | 17 |
| | Male | 9 |
| Age (years) | 18–30 | 1 |
| | 31–40 | 4 |
| | 41–50 | 5 |
| | 51–60 | 4 |
| | 61–70 | 3 |
| | 71–80 | 8 |
| | 80+ | 1 |
| Ethnic group | White | 26 |
| RMD diagnosis | Rheumatoid arthritis | 18 |
| | Psoriatic arthritis | 5 |
| | Inflammatory polyarthritis | 1 |
| | Polymyalgia rheumatica | 2 |
| How have your RMD symptoms been since the COVID-19 measures started? (reported at baseline) | Decreased | 3 |
| | Stayed the same | 9 |
| | Increased | 14 |

RMD, rheumatic and musculoskeletal diseases.

men and 17 women, with a mean age of 59 years (range: 29–83 years). Eighteen participants were diagnosed with RA, five with psoriatic arthritis, two with polymyalgia rheumatica and one with inflammatory polyarthritis. At the time of the baseline survey, 14 had reported increased symptoms, 9 stable symptoms and 3 decreased symptoms (table 3).

Most participants described themselves as shielding, although there was variation in interpretation and adherence to the formal guidelines. There were narratives of resilience, acceptance and adaptive coping strategies:

> I put up a little poster just to remind myself that I can only change what I am able to change. I just need to try and accept what I can't change. (RMD1 female aged 50–54 years)

Nonetheless, most highlighted challenges or stresses associated with the COVID-19 pandemic. In some, this was accompanied by reported worsening symptoms, including increased pain, stiffness and fatigue. Not all participants directly attributed their experience of symptoms to the COVID-19 context: '*My arthritis has flared up. I don't know if it's the change in situation or it was going to happen anyway*'. However, many spoke about reasons related to the pandemic which they perceived had underpinned symptom changes. Several speculated that a combination of these factors may have been at play. The reasons are outlined thematically below and summarised in table 4.

**Table 4** Key themes generated from the qualitative study – perceived underlying attributions for symptom change

| Experience of symptoms | Attributions from interview data | Representative quote |
|---|---|---|
| **Increased symptoms** | **Accessing healthcare**<br>Holding on to concerns | *I don't like to worry them, they've got enough to do … It's something that I feel I can manage until when this is all over hopefully I'll get an appointment to go in and I'll mention it then.* (RMD17 female aged 70–74 years) |
| | Postponement or cancellation of appointments, treatments and investigations | *I think that's something I feel like I've missed out on with the lockdown is, I would actually quite have liked to have some proper physio, especially on the knees and that. Because the swelling just doesn't seem to want to go, as much as I poke and prod it and things like that.* (RMD 10 female aged 50–59 years)<br>*It was planned for me to have an MRI scan on my neck, but then this all happened and it's just sort gone out the window… It's quite hard, day-to-day, with the pain levels in my neck, but under the current situation, I would rather put up with the pain than put myself at risk by going into a hospital.* (RMD20 male aged 50–54 years) |
| | Telephone appointments | *Where I've missed out because of the lockdown, is that although when I phoned the rheumatology department they've been really helpful, it's not like having proper management. So all of the questions that I would normally have asked, I don't seem to ask so much on the phone … It would be nice to say to somebody, 'This bit hurts, what can I do about it? Is this any good?' and things like that. But you need to have the physical and visual side of that.* (RMD 10 female aged 50–54 years) |
| | **Reduced physical activity**<br>Reduced planned exercise (including less opportunity or access to facilities to exercise) | *I think I'm so much less active, before all this I was going swimming three times a week, that was the best exercise for me really, because it didn't put as much pressure on my body as going to the gym, running or anything like that. And, I don't think I fully appreciated how much swimming helped, whereas now, obviously, I'm not doing anything and I am just only getting out of the house for a little bit once a week, I've just found everywhere has just stiffened up so much … It's a bit scary, really.* (RMD5 female aged 25–29 years)<br>*[My joints are] getting stiff because you don't go out much to get exercise … sitting down a lot makes your hips and your back ache and don't get up often where your chair is.* (RMD6 male aged 70–74 years) |
| | Reduced routine activity (eg, walking to shops and going to work) | *I've been finding, like, sitting at my desk all day for work, I do get quite sore and quite stiff after work, because I think maybe when I was in an office environment I would be going out at lunchtime or just walking around the office to talk to different people … generally if I'm not sat at my desk working, I suppose I'm sat on the sofa, I'm not really that mobile around the house.* (RMD5 female aged 25–29 years)<br>*[Pain and stiffness] have been a bit more apparent because I haven't been able to get out as much and use my joints. I find when I've been cooped up in a space and sitting down for a while, then obviously it affects you more. I can walk to work and back again and it frees things up a little bit. But obviously not being able to do that, it's not the same … it gets worse when you can't be as mobile as you could be.* (RMD12 male aged 35–39 years) |
| | **Working at home**<br>Lack of ergonomic work-space | *At work I have a Varidesk … so, I can stand up. I also have like a proper chair, like a medical chair … [Working at home] was horrendous because I was working off my lap on a laptop, bending over. So, my back and my shoulder was very, very painful.* (RMD2 female aged 35–39 years) |
| | Struggling with household tasks | *I used to have someone come on a Thursday and do my chores for me … So perhaps it's something like that as well, you know, sort of without the help there … I think with the loss of energy you get more of a hurting in the legs at night and that.* (RMD19 female aged 70–74 years) |
| | **Well-being**<br>Managing multiple roles within the home (Home-schooling, childcare and work) | *I probably went way too hard on myself, trying to plan lessons for [children]. You're also then working, you're also trying to run the house … completely overdid it … I do think the flare that I had was related to being in lockdown. I'm not sure I would have had it that bad if we were out of lockdown, because I would have been able to maybe adjust working life, the girls would have been at school, or my parents would have had them … It was definitely probably reactive from overdoing it … trying to juggle so many plates.* (RMD13 female aged 30–34 years) |
| | Anxiety or low mood associated with the pandemic and being in a clinically vulnerable group. | *I do struggle quite a lot with my mental health and I've found that my physical health has got so much worse, as well … it's definitely harder at the moment.* (RMD5 female aged 25–29 years)<br>*Currently, I've got swelling … and I've got pain in multiple areas. So, I would say I'm having a flare up … maybe it's the stress had an impact on my immune system.* (RMD2 female aged 35–39 years) |

Continued

**Table 4** Continued

| Experience of symptoms | Attributions from interview data | Representative quote |
|---|---|---|
| **Stable symptoms** | Maintaining exercise | *In terms of my arthritis and stuff, that's not been – touch wood – too bad. I've tried to keep on top of it by trying to stay as active as I can really … I've been probably doing 2K, 3K, 5Ks a week and then us, as a family unit, we try and go for a walk … so the pain has not been too bad.* (RMD14 male aged 35–39 years) |
| | Pre-existing stable or well-controlled condition | *I don't think there's a lot of difference really, it's quite well controlled anyway.* (RMD4 female aged 65–69 years) |
| **Decreased symptoms** | An opportunity for respite, improved well-being | *The arthritis was getting worse [and since lockdown] I've had no flare ups. It's all been very much under control, and I think that's partly being removed from the work situation but it's partly having that easier pace of life, and if my body needs to sleep there's not an alarm set, it just sleeps. I think before this I was feeling very tired and unwell … I now feel far calmer, physically rested, which helps no end.* (RMD21 female aged 45–49) |
| | More opportunity for exercise | *I'm actually feeling really good right now … I like to think it's the exercises … I've definitely done more daily exercises with my son and skipping and stuff and more bike rides. I like cycling anyway, but I've been on more long bike rides and probably been taking the dog for a walk probably more than I'd done before.* (RMD11 female aged 50–54 years) |

## Accessing healthcare

Many participants described 'holding on' to concerns about their disease because they did not want to trouble health services for fear of burdening the service, stating staff might be too busy due to COVID-19. Some had decided to miss blood tests due to anxiety around exposure to infection. Others were self-managing their symptoms:

> If I get any breakthrough pain where I'm really not happy, I can take two of my tramadol and … I take nortriptyline at night and I make sure that I'm getting a decent night's sleep. I've just been doing a lot of self-care. (RMD26 female aged 55–59)

Health concerns were also sometimes not reported due to postponed or cancelled appointments. There was a commonly expressed acceptance of this situation, feeling that others are 'worse off' at a time of crisis: '*They cancelled my appointments … under the circumstances, you accept it … and think well, there's people who are suffering and I'll just stand by*' (RMD6 male aged 70–74 years).

However, missing or postponed appointments created anxiety and a sense of detachment from healthcare support for some, compounding delayed advice seeking: '*I would have brought it up (at postponed appointment) and it would have been good to discuss those things, and you do worry*' (RMD13 female aged 30–34 years). Some had telephone appointments instead and although many preferred this, feeling it to be safer during the pandemic, some said that telephone appointments did not substitute entirely for face to face contact: '*They can't see the pain you're in; they can't feel your joints, they can't do any of that … does make everything so much trickier*' (RMD23 female aged 45–49). Others explained that they avoided raising what they thought of as smaller concerns during a telephone call. One person described a physiotherapy telephone appointment that they acknowledged had been helpful, but also said, '*[I]*

*need human intervention for my body*' (RMD10 female aged 55–59 years).

## Physical activity

Shielding instructions, lockdown and fear of catching COVID-19 meant many interviewed participants reported being much more sedentary and not leaving the house at all. The most common attribution for symptom change, including pain, stiffness and fatigue, was reduced activity: '*I think that's just lack of movement, the lack of being able to move about. I think, over time, your body just shuts down due to lack of mobility*' (RMD10 female aged 55–59 years).

Some discussed reduced motivation or limited opportunity to be able to engage in exercise: '*I used to go to the gym a lot, just to force myself. But they're all closed*' (RMD22 male aged 45–49). Reduced activity was also linked to disruption of everyday routine. This was more often discussed by those of working age who described ceasing a regular daily walk or cycle commute to work, or being on their feet in manual jobs and within the context of office-based work:

> Obviously, with me not working, then that's a huge change, I actually spend my life going from one of the departments to another … I would say my exercise routine is pretty rubbish, I'll be honest I have put some weight on during lockdown. That could be part of [worse symptoms], let's face it. (RMD1 female aged 50–54 years)

None reported having received specific advice about maintaining exercise for joint health at the time of interview, but a number were trying to substitute with exercise at home (eg, using a treadmill, online classes or walking round the garden). Exercising while avoiding contact with others was even more challenging without access to a private garden space:

I've got a long balcony here, it's about 100 yards long. So I should walk round there, but again, you've got to restrict that, because you never know if you're going to meet someone the other way. There isn't two metres to pass … you find you're just getting stiffer and stiffer. (RMD6 male aged 70–74 years)

### Home workspace and tasks at home

Several interviewees explained they were working from home in environments that were far from ideal for the maintenance of good joint health, for example, using a dining room chair and laptop rather than a desktop computer and office chair. Many envisaged home working for the foreseeable future. Levels of support and provision of equipment from employers was variable. Some employers had made swift efforts to prioritise provision of suitable equipment. However, others had not received this support, and one participant was hesitant to complain for fear of losing their job at a time of job insecurity:

I'm just sitting at the kitchen table with my laptop. [My employer] said they couldn't do anything because there were so many of us that weren't at work. I was just like well, I'll just get on with it then, because I kind of need my job. (RMD23 female aged 45–49 years)

In response to exacerbation of pain, some interviewees had purchased more ergonomic equipment themselves, while others felt they did not have the space to accommodate this kind of office set-up within their home.

A few participants who were shielding discussed no longer having access to assistance with household tasks that had adversely impacted on their physical or mental health. A few spoke of pacing out household activities but had not received professional advice on this.

### Well-being

Descriptions of low mood, isolation and boredom were common in the interviews. Many explained they were anxious about being in a clinically vulnerable group and also reported a sense of being left behind or forgotten about. Many expressed worries about the future, and some felt they would continue staying at home, regardless of relaxation of guidelines. A few said only a vaccine would make them feel 'safe'. Some made an explicit direct link with pain and fatigue symptoms, speculating, for example, that increased pain may be linked to stress surrounding their current situation. Others commented on the effect on their fatigue levels: '*I've felt more tired, but I think that is because I felt more anxious and I haven't slept so well … I'd say it's more linked to worrying about COVID-19*' (RMD11 female aged 50–54 years).

For some with caring responsibilities, increased demands and stress associated with managing multiple roles at home, working while schools were closed, was identified as having contributed to symptom flare-up. There was also additional stress around decision-making and risk assessment in the context of family life, managing dilemmas about protecting their own health against wider family well-being.

It's been exhausting, absolutely exhausting. I was getting [child] to try and do his work, but he really needs one-to-one because he gets bored so quickly. So that became a bit of a nightmare … and then working in the evening to kind of catch up … no way of, if my arthritis is really bad, of just resting. (RMD23 female aged 45–49 years)

### Improving or stable symptoms

In those who perceived no change in their condition, some attributed this to it having been in a stable, long-term state, for instance with disease that was already well-controlled by medication or conversely '*already bad*'. One participant who reported stable symptoms credited, in part, the support he had received from his employer prioritising him to receive appropriate office equipment for working at home. Others attributed their stable condition to maintaining routine, exercise and other protective coping behaviours: '*[My symptoms] seem to be about the same really. I know if I sit about I'll get stiff, so I do an exercise programme every day and try to keep myself moving around*' (RMD16 female aged 75–79 years).

A small number of those interviewed explained that their symptoms had markedly improved as a consequence of COVID-19 measures. One cited better opportunity and motivation to take more exercise such as walks or joining children for daily exercises. Another participant described benefiting from being at home and a slower pace of activity and that shielding had offered a period of respite during which they were better able to cope with their condition and prioritise self-care:

I'm furloughed … Having the time to just be at home, no one's expectations, my body has a chance to heal and find some balance … it makes you wonder. There's obviously no cure for RA at the moment but it does make you, you know, that emotional health is so key. (RMD21 female aged 45–49 years)

Online supplemental file 5 provides a summary of an interpretive synthesis of results, with interview findings, offering an explanatory insight into potential reasons for changes identified through the quantitative investigation.

### DISCUSSION

Within the survey cohort, we found that people with RMDs frequently reported a deterioration in pain and symptoms, in addition to greater social isolation, loneliness and reduced optimism to their circumstances. This was significantly worse for those aged 18–60 years in comparison with older participants. The qualitative findings suggest that more types of change to daily life were experienced by those in employment. Although some retired people reported reduced opportunity for exercise outside the home, they did not face the many competing

demands experienced by employed people and people with children at home.

## Relationship to previous evidence

Recently, Persiani *et al*[23] and Garrido-Cumbrera *et al*[24] reported their findings of the impact of the first COVID-19 pandemic 'lockdown' in with people with RMDs. These findings reflect those of our UK cohort, which illustrated a reduction in physical activity and increase in pain, particularly among people aged 50–70 years and in work.[23] and poor lifestyle habits negatively impacting on overall physical and mental health.[24] These findings may be attributed to a disruption in their normal working patterns, changes in their home environment and loss of social networks. Our qualitative interviews indicated that the enforced COVID-19 pandemic social restrictions led to stressful changes in working routine, the addition of extra roles within the home (eg, through home schooling) and deleterious workstation ergonomics due to lack of space and equipment. The latter point may be particularly important. Where occupational health requires workstation assessments for ergonomic safety, these were not practical in this phase of the pandemic. As people with pre-existing RMDs are particularly sensitive to the negative impacts of poor static postures, changes in workspace and working hours may have accounted for these reported physical responses.[25] As homeworking continues, employers should be aware of this challenge for employees with existing RMDs to help reduce home issues that could lead to symptom flares.

## Clinical implications

The findings have implications for clinical practice. First, the cohort we recruited largely demonstrated persistence in their worsening symptoms over time. As the qualitative research highlighted, reluctance to engage with health services, particularly during times of higher perceived risk, may mean that individuals are less likely to seek help for physical and mental health or only seek help when their symptoms become more severe, which may be more challenging to manage. Advertising and promoting services and how to access these may therefore be important strategies across primary and secondary care services to negate this issue. Second, due to changes in daily routine and increased symptoms, there may be greater disability within this population. Support to encourage resumption of physical activity routines and encouraging activity and engagement may be important for all.[26 27] Finally, these findings indicate that younger, working-aged people with RMDs were particularly affected by COVID-19 pandemic restrictions. Specific consideration of the impact on these individuals should be considered as they, potentially unexpectedly, may be at considerable risk of poorer musculoskeletal outcomes compared with older individuals. Employers should ensure that their workers, while working from home,

have suitable risk assessments and provision of equipment to minimise the risks of exacerbating musculoskeletal pain, particularly for those with pre-existing RMDs.

In the months after the data for this paper were collected (August 2020), some enforced changes have been lifted. However, for many people in the UK and internationally, measures to restrict social activity remain and their impacts are ongoing. The findings from this study therefore remain pertinent.[28] Furthermore, while some activities can be recommenced, such as attending hospital appointments and communal exercise, many people reported longer term concerns regarding their risk of COVID-19 and have refrained from resuming their normal routines. Additional support may be required to encourage individuals with the greatest anxiety to return to some of these social pursuits. This will be particularly important to avoid problems associated with social isolation, once longer term restrictions are lifted. Previous literature has demonstrated the association between social isolation and higher incidence of cardiovascular disease,[29 30] mental conditions,[31] dementia[32] and mortality.[33]

## Strengths and limitations

This study is a large, national cohort of individuals with RMDs. The mixed-methods approach provides novel and important real-time exploratory and explanatory findings to the changes in symptoms that this cohort reported during the initial 10 weeks of COVID-19 pandemic restrictions. Method triangulation[34] led to a more comprehensive understanding of the experiences of this specific group of people who may be considered particularly at risk of unintended consequences from social restrictions during the pandemic.

While these are key strengths, the study presents with three important limitations. First, while we can report the self-reported levels of disability and impairment, the sample were self-selecting when completing the survey. While the characteristics of this cohort are typical of those with RMDs,[35] responder bias may have an impact on the external validity of these findings. Second, while we are able to make assumptions based on a cohort that includes a range of different RMDs, we acknowledged that only a small proportion of the population were BAME, and we did not collect data to assess the respondents' level of social deprivation. Previous literature has highlighted these two factors to be potentially important in determining COVID-19 prognosis.[36] Including these groups may have provided a different description of the behaviours and perceptions of these individuals towards COVID-19 and the impact on their symptoms. Finally, the qualitative study recruited people from the NOAR cohort who had also completed the survey. These were all patients with inflammatory rheumatological conditions. It remains unclear whether the responses those

individuals included in our study can be applicable to non-inflammatory musculoskeletal conditions.

## CONCLUSIONS

People with RMDs frequently experienced deterioration in pain and symptoms when COVID-19 pandemic social restriction measures were enforced. As the restrictions continued over the following 10 weeks, levels of social isolation and loneliness increased and optimism decreased. These changes were greater among those aged 18–60 years when compared with older age groups. Close attention to those at risk through the promotion of physical activity, changing home-working practices and awareness of healthcare provision is important as social restrictions continue in the UK.

**Acknowledgements** The authors would like to thank the following organisations who assisted in distributing the survey across their networks: Pain Concern; Arthritis Action; National Rheumatoid Arthritis Society; National Ankylosing Spondylitis Society; Paget's Association; Parathyroid UK; The Health Policy Partnership; and Scope.

**Contributors** TOS, PB, JRD, KD, JRC, JT, MY, FN, SW, CN, LS and AJM researched the topic and devised the study; KD, JRC, TOS and MY performed the quantitative data collection; JRD performed the statistical analyses; PB and LB performed the qualitative data collection; PB and LB performed the qualitative analysis; SW provided Patient and Public Involvement support; TOS, PB, JRD, KD, JRC, JT, MY, FN, SW, CN, LS and AJM provided the first draft of the manuscript; TOS, PB, JRD, KD, JRC, JT, MY, FN, SW, CN, LS and AJM contributed equally to manuscript preparation; TOS, PB, JRD, KD, JRC, JT, MY, FN, SW, CN, LS and AJM approved the submitted manuscript. AJM and TOS obtained research funding. AJM acts a guarantor.

**Funding** Funding was obtained to support the conduct of this study from Action Arthritis (Norfolk) and the University of East Anglia.

**Competing interests** None declared.

**Patient consent for publication** Not required.

**Ethics approval** Ethical approval was obtained from the University of East Anglia's Faculty of Medicine and Health Science's Research Ethics Committee (RE: 2019/20–104 & 2019/20–105).

**Provenance and peer review** Not commissioned; externally peer reviewed.

**Data availability statement** Data are available on reasonable request.

**ORCID iDs**
Toby O Smith http://orcid.org/0000-0003-1673-2954
Jack R Dainty http://orcid.org/0000-0002-0056-1233
Linda Birt http://orcid.org/0000-0002-4527-4414
Felix Naughton http://orcid.org/0000-0001-9790-2796
Caitlin Notley http://orcid.org/0000-0003-0876-3304
Alex J MacGregor http://orcid.org/0000-0003-2163-2325

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
