## [Reviewer comments · BMJ Open]

ARTICLE DETAILS

TITLE (PROVISIONAL)	The impact of COVID-19 pandemic social restriction measures on people with rheumatic and musculoskeletal diseases in the UK: a mixed-methods study.
AUTHORS	Smith, Toby O.; Belderson, P; Dainty, Jack; Birt, Linda; Durrant, K; Chipping, JR; Tsigarides, J; Yates, M; Naughton, Felix; Werry, S; Notley, Caitlin; Shepstone, Lee; MacGregor, Alex

VERSION 1 – REVIEW

REVIEWER	De Meo, Daniele Sapienza University of Rome, Orthopaedics & Traumatology
REVIEW RETURNED	10-Mar-2021

GENERAL COMMENTS	General comments: This study explores the impact of Covid 19 lockdown on patients with musculoskeletal disorders. The study provides some new perspectives and presents useful information which could be of interest for the readers of BMJ Open. The research was conducted with rigorous methodology and presents significant results. Moreover, the narrative style appears extremely smooth to read. The results section is very well written and comprehensive but it's long. I found some parts of the discussion that might be enhanced. I suggest the Editor to accept the article after some minor changes and re-evaluation by reviewer. I ask that the Authors specifically address each of my comments in their response. Title: ok. Abstract: It is very well written, easy to understand and coherent. Introduction: It frames the main problems of this topic and identifies the purpose of the study. Methods: correctly explains the research conducted and makes it reproducible. Results: The results are interesting. Anyway, the analysis of the qualitative study outweighs the analysis of the survey results. The Authors might consider reducing the space dedicated to qualitative results giving more space to survey's results. Discussion: The discussion covers almost all the aspects dealt with in the results but could benefit from more comparison with recently published work in the literature. Here some examples. - Line 4, page 11: What do the authors mean by "Relationship to Previous Evidence"? If they refer to previously published literature on the subject, they should think about expanding their references. The authors might consider comparing their results with the following recently published article, which examines the same aspects of the authors' work with the additional inclusion of patients with post-traumatic orthopaedic conditions: Persiani P, De Meo D, Giannini E, Calogero V, Speziale Varsamis T, Cavallo AU, Martini L, Cera G, Coluzzi F, Villani C. The
---

	Aftermath of COVID-19 Lockdown on Daily Life Activities in Orthopaedic Patients. J Pain Res. 2021;14:575-583 https://doi.org/10.2147/JPR.S285814 Please also correct the highlights bullet points considering the existence of this paper ("This is the first study to detail the impact of COVID-19 social restrictions on the lived experiences of people with chronic musculoskeletal diseases."). - It might be interesting for readers to understand how the telephone visits were conducted, in order to better understand the patients' response in the manuscript. About the role of the telemedicine, the Authors could benefit from these two articles that highlight interesting new methods of orthopaedic virtual examination: Tanaka MJ, Oh LS, Martin SD, Berkson EM. Telemedicine in the Era of COVID-19: The Virtual Orthopaedic Examination. J Bone Joint Surg Am. 2020 Jun 17;102(12):e57. doi: 10.2106/JBJS.20.00609. Wahezi SE, Duarte RA, Yerra S, Thomas MA, Pujar B, Sehgal N, Argoff C, Manchikanti L, Gonzalez D, Jain R, Kim CH, Hossack M, Senthelal S, Jain A, Leo N, Shaparin N, Wong D, Wong A, Nguyen K, Singh JR, Grieco G, Patel A, Kinon MD, Kaye AD. Telemedicine During COVID-19 and Beyond: A Practical Guide and Best Practices Multidisciplinary Approach for the Orthopedic and Neurologic Pain Physical Examination. Pain Physician. 2020 Aug;23(4S):S205-S238. Conclusion: Ok. Images and Tables: ok Reference: ok
--	--

REVIEWER	Garrido-Cumbrera, Marco University of Seville
REVIEW RETURNED	20-Mar-2021

GENERAL COMMENTS	GENERAL COMMENTS  - This article assesses the impact of the restrictions imposed by the COVID-19 pandemic using information from a national online survey (between April and August 2020) on a sample of 708 people with chronic musculoskeletal conditions and through a qualitative interview-based study both in the UK. - Overall it is difficult to follow as the difference between the information collected through the survey (quantitative) and the information collected through interviews (qualitative) is not clear. It is recommended that a clear distinction be made between these two parts and that some relationship be made between the two types of information in order to understand whether they are in line with each other. - The results over time collected through the longitudinal survey are not explained. Is it possible that it is a survey collected over a period of time, but without patients participating several times? It is important to describe whether it is a longitudinal or a cross-sectional study, which is not clear. MAJOR CHANGES  - This study refers to the sample as people with musculoskeletal diseases when it would be more appropriate to indicate rheumatic and musculoskeletal diseases (RMDs) which is the term most used internationally by EULAR, ACR or BSR among others. In addition, "Rheumatology" is included as Keywords and "chronic musculoskeletal diseases" is not mentioned there. - STRENGTHS AND LIMITATIONS OF THIS STUDY (lines 49 and 50): I disagree with the idea that this is the first study done on
--

	the experiences of people with chronic diseases from their perspective. There is a study that has evaluated the experience of patients with RMDs during the pandemic in several European countries, including the UK, called REUMAVID so this sentence should be deleted.  - METHOD (lines 53-55): include the question that was used for patients to self-report their disease (chronic and/or musculoskeletal). - METHOD (lines 58-59): further explain what the qualitative study consisted of, describing the specific questions used. - METHOD (LINES 11-25): In the Data Collection section, various scales such as the CLINHAQ or the Lubben Social Network Scale are reported, but the results obtained using these scales are not reported. - TABLE 2 (LINES 6-40): it is difficult to understand when there are significant differences between the groups compared. For example, the percentage of people who changed their medication was exactly the same between those whose symptoms were reduced and those whose symptoms were increased. However, the p-value shows that these differences were significant. - TABLE 3 (lines 5-24) shows how the proportion of patients with RA was 18 versus only 8 for the other 3 RMDs. This should be mentioned in the study limitations section. MINOR CHANGES  - When mentioning Ankylosing Spondylitis (AS) and not using the current term Axial Spondyloarthritis (axSpA) which also includes AS would have been more appropriated. Although this should not be changed now as it is understood that this is how the data was collected. In any case, the list of diseases included in the survey should be included in the methodology section (and not only at the table). - When referring to the pandemic, the word pandemic should be included next to COVID-19 (ie COVID-19 pandemic) and not simply COVID-19, especially at the beginning of the manuscript. - Ethical Considerations (lines 19 and 20): The Ethical approval refer to 2019 while the study was conducted in 2020. Maybe the 2019 refer to the code of the EC but otherwise it should be corrected. - In general, the article uses underlining, bold or capital letters for headings indiscriminately, making it difficult to understand the organisation of the article. - A typo in the following sentence "....may have provided different a different description...".
--	--

VERSION 1 – AUTHOR RESPONSE

REVIEWER 1

Comment: This study explores the impact of Covid 19 lockdown on patients with musculoskeletal disorders. The study provides some new perspectives and presents useful information which could be of interest for the readers of BMJ Open. The research was conducted with rigorous methodology and presents significant results. Moreover, the narrative style appears extremely smooth to read. The results section is very well written and comprehensive but it's long. I found some parts of the discussion that might be enhanced. I suggest the Editor to accept the article after some minor changes and re-evaluation by reviewer. I ask that the Authors specifically address each of my comments in their response.

Response: Thank you for your kind words. We have addressed the points raised by the reviewer and itemised these below.

Comment: Title: ok.

Response: No amendment required.

Comment: Abstract: It is very well written, easy to understand and coherent.

Response: No amendment required.

Comment: Abstract: Introduction: It frames the main problems of this topic and identifies the purpose of the study.

Response: No amendment required.

Comment: Abstract: Methods: correctly explains the research conducted and makes it reproducible.

Response: No amendment required.

Comment: Abstract: Results: The results are interesting. Anyway, the analysis of the qualitative study outweighs the analysis of the survey results. The Authors might consider reducing the space dedicated to qualitative results giving more space to survey's results.

Response: Where the meaning is not lost, we have reduced text within illustrative quotes (**Results, Qualitative study, Accessing health care, Quote 1; Physical activity, Quote 2, 3 and 4; Home work-space and tasks at home, Quote 1**). Some quotes have also been removed entirely from **Table 4**. In order for the reader to understand findings we feel much of the interpretive text needs to be retained but have again edited to remove words wherever possible.

Comment: Abstract: Discussion: The discussion covers almost all the aspects dealt with in the results but could benefit from more comparison with recently published work in the literature. Here some examples.

Response: Thank you. We have used the main-paper to contrast the results to recent published work, as we were constrained by word-count in the Abstract. These are presented in the main-paper (**Discussion, Paragraph 2, Lines 1-4**).

Comment: Line 4, page 11: What do the authors mean by "Relationship to Previous Evidence"? If they refer to previously published literature on the subject, they should think about expanding their references. The authors might consider comparing their results with the following recently published article, which examines the same aspects of the authors' work with the additional inclusion of patients with post-traumatic orthopaedic conditions:

Persiani P, De Meo D, Giannini E, Calogero V, Speciale Varsamis T, Cavallo AU, Martini L, Cera G, Coluzzi F, Villani C. The Aftermath of COVID-19 Lockdown on Daily Life Activities in Orthopaedic Patients. *J Pain Res.* 2021;14:575-583

Response: Thank you for highlighting this paper. This was not available at the time of preparing this manuscript. We have taken this opportunity to incorporate the findings of this paper in this revised manuscript (**Discussion, Paragraph 2, Lines 1-4**).

Comment: Please also correct the highlights bullet points considering the existence of this paper ("This is the first study to detail the impact of COVID-19 social restrictions on the lived experiences of people with chronic musculoskeletal diseases.").

Response: As far as the authors are aware, this is the first study to apply mixed-methods and incorporate an in-depth qualitative research element to allow a depth of contextual understanding of this research question. The authors believe that the RHEUMAVID study, whilst incredibly important, is a survey and does not aim to investigate qualitative research questions. Nonetheless, as

recommended by the reviewer, we have modified this summary point (**Strengths and Limitations of the Study, Bullet Point 1**).

Comment: It might be interesting for readers to understand how the telephone visits were conducted, in order to better understand the patients' response in the manuscript. About the role of the telemedicine, the Authors could benefit from these two articles that highlight interesting new methods of orthopaedic virtual examination:

Tanaka MJ, Oh LS, Martin SD, Berkson EM. Telemedicine in the Era of COVID-19: The Virtual Orthopaedic Examination. J Bone Joint Surg Am. 2020 Jun 17;102(12):e57. doi: 10.2106/JBJS.20.00609.

Wahezi SE, Duarte RA, Yerra S, Thomas MA, Pujar B, Sehgal N, Argoff C, Manchikanti L, Gonzalez D, Jain R, Kim CH, Hossack M, Senthelal S, Jain A, Leo N, Shaparin N, Wong D, Wong A, Nguyen K, Singh JR, Grieco G, Patel A, Kinon MD, Kaye AD. Telemedicine During COVID-19 and Beyond: A Practical Guide and Best Practices Multidisciplinary Approach for the Orthopedic and Neurologic Pain Physical Examination. Pain Physician. 2020 Aug;23(4S):S205-S238.

Response: Thank you for raising this point. The purpose of the survey was to explore symptoms and lived experiences of this population. We did not use telephone visits to follow-up patients clinically. We feel that this is outside the remit of this paper, and has therefore not incorporated this into the revised manuscript. If the review and editor feel strongly about this decision, we would be happy to reconsider.

Comment: Conclusion: Ok.

Response: No amendment required.

Comment: Images and Tables: ok

Response: No amendment required.

Comment: Reference: ok

Response: No amendment required.

REVIEWER 2

Comment: This article assesses the impact of the restrictions imposed by the COVID-19 pandemic using information from a national online survey (between April and August 2020) on a sample of 708 people with chronic musculoskeletal conditions and through a qualitative interview-based study both in the UK. Overall it is difficult to follow as the difference between the information collected through the survey (quantitative) and the information collected through interviews (qualitative) is not clear. It is recommended that a clear distinction be made between these two parts and that some relationship be made between the two types of information in order to understand whether they are in line with each other.

Response: The team have reviewed the text in order to ensure this point is raised. This is frequently a challenge in reporting a mixed-methods study but, on reflection, we feel that the paper appropriately synthesis the two parts rather than making a clear distinction, which is not the intention of mixed-methods research. On reflection, we feel that there is currently a very clear distinction is made in the Results section (**Quantitative and Qualitative Study headings**). However, in this revised paper we have provided greater signposting in the Discussion as to which element of the study we are drawing on during the interpretation of the text (**Discussion, Paragraph 1, Line 1**).

The reviewer makes an important point regarding “some relationship be made between the two types of information in order to understand”. This is the exact intention of the figure in **Supplementary File 5**, to provide an interpretative synthesis of results from both strands. **Table 4** is also designed to do this – presenting attributions from interview data which provide contextual understanding for the direction of symptom change (increased, stable, decreased) reported in the survey. This has been

clarified in the text (**Methods and Analysis, Qualitative Study, Data Analysis, Paragraph 2, Lines 2-4**).

Comment: The results over time collected through the longitudinal survey are not explained. Is it possible that it is a survey collected over a period of time, but without patients participating several times? It is important to describe whether it is a longitudinal or a cross-sectional study, which is not clear.

Response: Thank you for raising this important point. The reviewer is correct in that this is a longitudinal study. However approximately 25-30% of the participants dropped-out after the baseline questionnaire. We have added text in the Results section (**Results, Follow-up Assessment, Paragraph 1, Lines 1-5**) to clarify the numbers in the study as time progressed. We have also added a Table (**Supplementary file 4**) to compare the characteristics of the participants at baseline to those who also completed the week=10 questionnaire and shows no important differences between the two groups.

Comment: This study refers to the sample as people with musculoskeletal diseases when it would be more appropriate to indicate rheumatic and musculoskeletal diseases (RMDs) which is the term most used internationally by EULAR, ACR or BSR among others. In addition, "Rheumatology" is included as Keywords and "chronic musculoskeletal diseases" is not mentioned there.

Response: We have made this amendment as recommended throughout (**Abstract, Objectives, Line 2; Abstract, Participants, Line 1, Abstract, Results, Line 4-5; Abstract, Conclusions Line 1; Strengths and Limitations, Bullet Point 1; Introduction, Paragraph 1, Lines 1-2; Introduction, Paragraph 3, Lines 1 & 5; Introduction, Paragraph 4, Line 2 & 4; Methods and Analysis, Cohort and Recruitment, Paragraph 1, Line 2; Methods and Analysis, Qualitative Study, Paragraph 1, Line 2; Results, Survey Results, Paragraph 2, Line 3; Discussion, Paragraph 1, Line 1; Discussion, Paragraph 2, Line 8 & 11; Discussion, Paragraph 3, Line 11 & 16; Discussion, Paragraph 4, Line 1; Discussion, Paragraph 5, Lines 4 & 6; Conclusions, Line 1**). We have also updated our keywords to reflect the reviewer's suggestion (**Abstract, Keywords**).

Comment: STRENGTHS AND LIMITATIONS OF THIS STUDY (lines 49 and 50): I disagree with the idea that this is the first study done on the experiences of people with chronic diseases from their perspective. There is a study that has evaluated the experience of patients with RMDs during the pandemic in several European countries, including the UK, called REUMAVID so this sentence should be deleted.

Response: As recommended, we have modified this summary point (**Strengths and Limitations of the Study, Bullet Point 1**).

Comment: METHOD (lines 53-55): include the question that was used for patients to self-report their disease (chronic and/or musculoskeletal).

Response: For clarity, and as recommended by the Editor, we have included the survey as **Supplementary File 1**. This is acknowledged in the text (**Methods and Analysis, Survey Study, Data Collection, Paragraph 1, Lines 2-3**). Through this, the reader will be able to ascertain the question used to report their RMD in addition to being able to understand all other questions which were posed.

Comment: METHOD (lines 58-59): further explain what the qualitative study consisted of, describing the specific questions used.

Response: We have added the interview topic guide as **Supplementary File 3**.

Comment: METHOD (LINES 11-25): In the Data Collection section, various scales such as the CLINHAQ or the Lubben Social Network Scale are reported, but the results obtained using these scales are not reported.

Response: These data are presented in **Table 1**. We have therefore kept the reporting of these measures in the Methods section.

Comment: TABLE 2 (LINES 6-40): it is difficult to understand when there are significant differences between the groups compared. For example, the percentage of people who changed their medication was exactly the same between those whose symptoms were reduced and those whose symptoms were increased. However, the p-value shows that these differences were significant.

Response: We are grateful to the reviewer to pointing this out and agree that **Table 2** as formatted was difficult to interpret.

To make things clearer, we have changed the column percentages to row percentages. We have applied chi squared tests to address whether the reported change in symptoms at baseline varied for each explanatory variable (age, gender, diagnosis, situation, access to medication, changing medication, consulting health profession and physical activity). We feel this makes the Table simpler to interpret. Focusing on the example the reviewer has highlighted relating to change in medication, the chi square test indicates that change in symptoms levels since lockdown differed significantly between those who changed their medication and those who did not. Inspecting the Table, 71% of those who had changed their medication reported increased symptoms compared with 49% among those had not changed their medication.

We have altered the Methods section (**Methods and Analysis, Survey study, Data analysis, Paragraph 1, Lines 3-6**) to clarify what is being compared and the statistical approach. For further clarity, we have added confidence intervals to the proportions of those reporting increased, same or decreased symptoms at baseline (**Results, Survey Results, Paragraph 2**), and no longer include the proportional chance criterion test included in the earlier draft.

Comment: TABLE 3 (lines 5-24) shows how the proportion of patients with RA was 18 versus only 8 for the other 3 RMDs. This should be mentioned in the study limitations section.

Response: We have highlighted this as a study limitation in the Discussion (**Discussion, Paragraph 6, Lines 11-14**).

Comment: MINOR CHANGES: When mentioning Ankylosing Spondylitis (AS) and not using the current term Axial Spondyloarthritis (axSpA) which also includes AS would have been more appropriated. Although this should not be changed now as it is understood that this is how the data was collected. In any case, the list of diseases included in the survey should be included in the methodology section (and not only at the table).

Response: We agree that it would be inappropriate to change the reporting of AS to axSpA at this stage given this was the interpretation of the respondents in the survey. However, as recommended, we have included the reporting of disease in the Methods section (**Methods and Analysis, Survey Study, Data Collection, Paragraph 2, Bullet Point 1**).

Comment: When referring to the pandemic, the word pandemic should be included next to COVID-19 (ie COVID-19 pandemic) and not simply COVID-19, especially at the beginning of the manuscript.

Response: As recommended by the reviewer, we have amended this throughout the paper (**Title; Abstract, Objectives, Line 1; Abstract, Results, Line 2; Abstract, Conclusions, Line 2; Introduction, Paragraph 3, Line 7; Introduction, Paragraph 4, Line 1; Methods and Analysis, Qualitative Study, Paragraph 1; Results, Survey Results, Paragraph 2, Line 2; Discussion, Paragraph 2, Line 3; Discussion, Paragraph 3, Line 11; Discussion, Paragraph 4, Line 6; Discussion Paragraph 5, Line 3; Conclusions, Line 2**).

Comment: Ethical Considerations (lines 19 and 20): The Ethical approval refer to 2019 while the study was conducted in 2020. Maybe the 2019 refer to the code of the EC but otherwise it should be corrected.

Response: The Ethical Committee reference number is correctly reported.

Comment: In general, the article uses underlining, bold or capital letters for headings indiscriminately, making it difficult to understand the organisation of the article.

Response: We have revised the text throughout, correcting the headings in all sections as recommended (**throughout Methods section, Results section, Discussion**).

Comment: A typo in the following sentence "...may have provided different a different description...".

Response: Thank you for highlighting this. We have made this correction (**Discussion, Paragraph 6, Line 9**).

VERSION 2 – REVIEW

REVIEWER	Garrido-Cumbrera, Marco University of Seville
REVIEW RETURNED	06-May-2021

GENERAL COMMENTS	GENERAL COMMENTS The authors have implemented correctly my suggestions. The manuscript is much improved and more appealing to read. MINOR CHANGES - In the abstract section, under PARTICIPANTS please remove "chronic" and just leave "People in the UK with RMDs". The term "chronic" is repeated before RMDs in other part where should be removed as well. - In the abstract section, under the results part replace "703 people" by "703 people with RMDs" or "703 patients with RMDs". - In the keywords, replace "coronavirus" by "COVID-19 pandemic" and also include "lockdown". - At the TABLE AND FIGURE LEGENDS page in page no. 54/67 or no. 13 please add a point at the end of each of the sentences. - I don't think it is appropriate this acronym MSK for musculoskeletal. This should be revised across the whole manuscript. For instance, table 2 refers to musculoskeletal (MSK) rather than to RMDs. - I recommend to quote the following related study: Assessment of impact of the COVID-19 pandemic from the perspective of patients with rheumatic and musculoskeletal diseases in Europe: results from the REUMAVID study (phase 1) https://rmdopen.bmj.com/content/7/1/e001546
--

VERSION 2 – AUTHOR RESPONSE

Reviewer 1

No comments

Reviewer 2

Comment: The authors have implemented correctly my suggestions. The manuscript is much improved and more appealing to read.

Response: Many thanks for your kind words. We are pleased you agreed with our amendments. We have made the minor changes you have now suggested and itemised these below. Thank you in advance.

Comment: MINOR CHANGES: In the abstract section, under PARTICIPANTS please remove “chronic” and just leave “People in the UK with RMDs”. The term “chronic” is repeated before RMDs in other part where should be removed as well.

Response: This has been addressed as suggested (**Abstract, Participants, Line 1; Abstract, Results, Line 4; Abstract, Conclusions, Line 1; Keywords; Introduction, Paragraph 1, Line 2; Introduction, Paragraph 2, Line 5; Introduction, Paragraph 4, Line 2; Methods and Analysis, Qualitative Study, Paragraph 1, Line 2; Discussion, Paragraph 1, Line 1; Conclusion, Line 1**).

Comment: In the abstract section, under the results part replace “703 people” by “703 people with RMDs” or “703 patients with RMDs”.

Response: Addressed as suggested (**Abstract, Results, Line 1**).

Comment: In the keywords, replace “coronavirus” by “COVID-19 pandemic” and include “lockdown”.

Response: Addressed as suggested (**Keywords**)

Comment: At the TABLE AND FIGURE LEGENDS page in page no. 54/67 or no. 13 please add a point at the end of each of the sentences.

Response: Added full stop at the end of Legends as suggested (**Legends for Supplementary File 1; Supplementary File 3; Supplementary File 4; Supplementary File 5**).

Comment: I don’t think it is appropriate this acronym MSK for musculoskeletal. This should be revised across the whole manuscript. For instance, table 2 refers to musculoskeletal (MSK) rather than to RMDs.

Response: The abbreviation RMD has been used to replace MSK throughout the paper (**Results; Qualitative Study, RMD coding for participant responses throughout; Table 1; Table 2; Table 3; Table 4**).

Comment: I recommend to quote the following related study: Assessment of impact of the COVID-19 pandemic from the perspective of patients with rheumatic and musculoskeletal diseases in Europe: results from the REUMAVID study (phase 1)

<https://eur01.safelinks.protection.outlook.com/?url=https%3A%2F%2Frmddopen.bmj.com%2Fcontent%2F7%2F1%2F001546&data=04%7C01%7Ctoby.smith%40uea.ac.uk%7C5a221060d3ac42773fe308d915283b74%7C65f8795ba3d43518a070865e5d8f090%7C0%7C0%7C637564084813515126%7CUnknown%7CTWFPbGZsb3d8eyJWljoIMC4wLjAwMDAiLCJQljoIV2luMzliLCJBTiI6lk1haWwiLCJXVCI6Mn0%3D%7C1000&data=CnjFf5QI%2FCHySbQjV9j0GKHpVuGxyeu5iHBIDYqFR4%3D&reserved=0>

Response: we have incorporated the findings from this valuable paper in our Discussion (**Discussion, Paragraph 2, Lines 1-5; Reference 24**). Thank you for highlighting this.